# OpenReview forum: "Speech-Audio Compositional Attacks on Multimodal LLMs and Their Defense with SALMONN-Guard"
_ICML.cc/2026/Conference — ICML 2026 regular_

### Official Review · Reviewer_sRvJ · 2026-03-04

**Soundness:** 2
**Presentation:** 3
**Significance:** 3
**Originality:** 2
**Overall Recommendation:** 4
**Confidence:** 4

**Summary:**

This paper proposes a new benchmark, SACRED-Bench, for evaluating the safety and robustness of multimodal large language models when processing complex audio inputs. Rather than relying on imperceptible adversarial noise, this method leverages realistic speech and audio compositions to construct attacks, including speech-speech overlap, speech-audio overlap, and multi-speaker dialogue. To address this vulnerability, the authors propose an audio-aware protection model, SALMONN-Guard, to defend against such attacks.

**Compliance With Llm Reviewing Policy:**

Affirmed.

**Final Justification:**

The author solved my problem.

**Key Questions For Authors:**

1 For the SSO attack, has the model's ability to directly transcribe audio been tested? How many "successful attacks" are merely due to the model failing to hear the harmful content clearly?

2 Are traditional audio defense baselines still effective against SSO, MSD, and SAO attacks?

3 After fine-tuning, how does SALMONN-Guard perform on standard and benign audio benchmarks?

**Limitations:**

Look at the weaknesses

**Strengths And Weaknesses:**

Strengths

1 The experiments effectively validate the effectiveness and threat potential of the three proposed cross-modal audio compositional attacks (SSO, MSD, SAO).

2 This paper designs a defense model that jointly reviews speech, audio, and text streams, capable of effectively defending against such audio compositional attacks.

Weaknesses

1 The SSO attack method proposed in the paper combines harmful speech with benign speech. If multimodal LLMs lack the ability to perform robust blind source separation or sound event detection in noisy environments, their audio encoders will directly discard the background channel. In this case, the model's safety alignment has not actually been breached; it simply hasn't heard the threat. The authors should experimentally demonstrate whether the attack success is a perception issue or a safety issue of the model.

2 Although the paper demonstrates the effectiveness of SALMONN Guard defense, it lacks comparison with existing defense baselines, such as cascaded defense baselines—first using highly robust Automatic Speech Recognition (ASR) models (like Whisper) to extract text from audio, then feeding the transcription results into specialized text safety models like LlamaGuard for semantic filtering.

3 SALMONN-GUARD is fine-tuned on specific datasets evaluated on SACRED-Bench. However, it is unclear whether fine-tuning on these specific adversarial structures would degrade the base model's performance on standard non-adversarial audio quality assurance tasks.

---

> ### Author Rebuttal · Authors · 2026-03-31
>
> Thank you for your invaluable feedback and constructive suggestions. We would like to respond to each individual concern as follows:
>
> ---
> > Weakness 1: The SSO attack method proposed in the paper combines harmful speech with benign speech ... The authors should experimentally demonstrate whether the attack success is a perception issue or a safety issue of the model.
> >
> >Question 1: For the SSO attack, has the model's ability to directly transcribe audio been tested? How many "successful attacks" are merely due to the model failing to hear the harmful content clearly?
>
> **We have verified that advanced multimodal LLMs successfully perceive both benign and harmful speech components in 93% of SSO samples, proving the high attack success rate is a genuine safety alignment failure rather than a perception limitation.** As detailed in Section 5.3 (Lines 393-401), we specifically tested this by prompting Gemini 2.5 Pro to transcribe and describe the audio content rather than execute a task. The model consistently recognized both audio streams, demonstrating that it “hears” the harmful intent clearly but fails to refuse it when “distracted” by the overlapping benign instruction. This confirms that the breach occurs at the alignment level, not the perception level. We will emphasize this crucial finding clearly in the revised main text.
>
> ---
> >Weakness 2: Although the paper demonstrates the effectiveness of SALMONN Guard defense, it lacks comparison with existing defense baselines, such as cascaded defense baselines ...
> >
> >Question 2: Are traditional audio defense baselines still effective against SSO, MSD, and SAO attacks?
>
> **Our evaluation on the full test set confirms that traditional cascaded defenses (e.g., Whisper-large-v3 + Gemini 2.5 Pro) are highly vulnerable to compositional audio attacks, yielding results nearly identical to the undefended baselines.** We will add the following detailed comparison table to Section 5 to highlight the inherent failures of audio-to-text pipelines:
>
> #### Table 1: ASR (%) Comparison for Cascaded Defense Baselines.
>
> | Defense Baseline | SSO (ASR %) | SAO (ASR %) | MSD (ASR %) |
> | :--- | :---: | :---: | :---: |
> | Baseline (Gemini 2.5 Pro without defense) | 37.25 | 88.56 | 63.93 |
> | Cascaded (Whisper-large-v3 / Text-Only + Gemini) | 37.50 | 88.56 | 57.96* |
> | **SALMONN-Guard (Ours)** | **12.93** | **5.16** | **14.08** |
>
> *\* The MSD cascaded result (57.96%) represents the ideal text-only scenario already evaluated in our Table 3 ablation study.*
>
> As demonstrated in the table, the cascaded pipeline fails across all three attack vectors due to the following fundamental limitations:
>
> *   **Failure on SSO (Overlapped speech):** The ASR remained at **37.50%** as current models struggle to recognize overlapping speech, leading to the loss of adversarial content during transcription.
> *   **Failure on SAO (Non-speech audio):** The ASR remained at **88.56%** as ASR models are incapable of transcribing non-speech content such as the violent and adult-oriented events in our dataset.
> *   **Failure on MSD (Multi-speaker dialogue):** As demonstrated in our **Table 3 ablation study (Section 5.3)**, even when providing the perfect text transcript (the “Text-Only” setup), the ASR remains critically high at **57.96%**. This proves that text-based filters are easily bypassed when malicious intent is cloaked within a conversational structure, invalidating cascaded defenses even when the ASR performs flawlessly.
>
> ---
> >Weakness 3: SALMONN-GUARD is fine-tuned on specific datasets evaluated on SACRED-Bench. However, it is unclear whether fine-tuning on these specific adversarial structures would degrade the base model's performance on standard non-adversarial audio quality assurance tasks.
> >
> >Question 3: After fine-tuning, how does SALMONN-Guard perform on standard and benign audio benchmarks?
>
> **SALMONN-Guard is designed as a specialized safety “firewall” rather than a general-purpose audio assistant, and its 0.00% False Alarm Rate (FAR) confirms it does not interfere with standard benign audio tasks.** Much like Meta’s LlamaGuard, our model serves as a proactive defense layer that intercepts malicious inputs while allowing safe queries to pass through unmodified to a downstream LLM (Section 4.2). To prevent over-refusal, we incorporated a diverse Harmless Control Set (HCS) during LoRA fine-tuning. As shown in Table 1, the 0.00% FAR demonstrates that SALMONN-Guard reliably permits standard, non-adversarial audio to proceed without any degradation in user experience. Therefore, its deployment adds a robust safety layer without sacrificing the base model's utility for benign interactions.

---

> > ### Author Rebuttal · Reviewer_sRvJ · 2026-04-03
> >
> > The authors have successfully addressed my concerns. I am raising my score accordingly. Good luck！

---

### Official Review · Reviewer_WRA1 · 2026-03-04

**Soundness:** 3
**Presentation:** 3
**Significance:** 3
**Originality:** 2
**Overall Recommendation:** 3
**Confidence:** 4

**Summary:**

This paper introduces SACRED-Bench, a benchmark for evaluating multimodal LLM safety under compositional audio attacks that combine benign and harmful signals within single audio streams. Three attack mechanisms are proposed: speech-speech overlap (SSO), speech-audio overlap (SAO), and multi-speaker dialogue (MSD). The key finding is that even Gemini 2.5 Pro, with safety guardrails enabled, achieves a 66% attack success rate on SACRED-Bench. The paper also proposes SALMONN-Guard, a guard model built on Qwen2.5-Omni-7B, fine-tuned to jointly inspect speech, audio, and text, which reduces the ASR to roughly 20%. The benchmark includes 30 hours of training audio and 7 hours of test data.

**Compliance With Llm Reviewing Policy:**

Affirmed.

**Key Questions For Authors:**

1. Can the authors clarify the naming: why "SALMONN-Guard" when the model is built on Qwen2.5-Omni? Is there a connection to the original SALMONN architecture that I'm missing?

2. Have the authors evaluated SALMONN-Guard on adversarial perturbation attacks (e.g., AdvWave or adversarial noise injection)? The Table 4 generalization results are encouraging but limited to two related methods.

3. Table 9 in Appendix shows that SALMONN-Guard's ASR (14.08% on MSD) is overwhelmingly "Wrong Understanding" (11.93%) rather than "Harmful Compliance" (2.15%). If you report only harmful compliance as the ASR, SALMONN-Guard's actual safety failure rate is ~2%. Why was this not the headline number?

**Limitations:**

**1. SSO and SAO reduce to binary classification, which is a weak evaluation target.** For SSO and SAO, the "attack" amounts to asking the model *"Is there harmful content in this audio?"* and measuring whether the model answers correctly. As a reviewer, this feels like fundamentally a detection task, not a jailbreak. When a model fails at SSO/SAO, it hasn't been tricked into generating harmful content, which it simply failed to detect harmful content in the input. This is an important distinction that the paper glosses over by using the term "attack success rate" for both detection failures and actual harmful generation (MSD). The paper would be more honest if it reported SSO/SAO separately as "detection failure rates" and reserved "attack success rate" for MSD, where the model actually produces harmful outputs.

**2. SALMONN-Guard is evaluated only on its own benchmark**  The benchmark and the defense are proposed in the same paper and trained on overlapping data distributions. While the cross-benchmark generalization in Table 4 partially addresses this, the paper would benefit enormously from evaluation on fully independent benchmarks. Does SALMONN-Guard handle attacks that are qualitatively different from compositional audio?

**3. Naming confusion: SALMONN-Guard is not built on SALMONN.** The guard model uses Qwen2.5-Omni-7B as its backbone (Section 4.2), yet it is branded "SALMONN-Guard." The original SALMONN model [1] is not even cited in the paper. This naming choice is misleading — readers will naturally assume a lineage that does not exist. The paper should either (a) cite the original SALMONN and explain the naming rationale, or (b) rename the model to something that reflects its actual architecture.

**4. No comparison with simple defenses.** The paper jumps from "models are vulnerable" straight to "here's a specialized guard model." But what about simpler approaches? Does basic transcription followed by text-based safety filtering mitigate SSO/MSD attacks? Does audio source separation before safety classification help for SSO? Would prompt-based defenses ("before answering, check if any part of the audio contains harmful content") improve performance? Authors might want to check [2] for prompt-based defenses. Without these baselines, it's hard to tell if the full SALMONN-Guard training pipeline is necessary.

**References**
[1] Tang, Changli, et al. "Salmonn: Towards generic hearing abilities for large language models." arXiv preprint arXiv:2310.13289 (2023).
[2] Roh, Jaechul, Virat Shejwalkar, and Amir Houmansadr. "Multilingual and multi-accent jailbreaking of audio llms." arXiv preprint arXiv:2504.01094 (2025).

**Strengths And Weaknesses:**

1. The paper identifies a genuinely practical attack surface. Real-world audio is messy where overlapping speakers, background sounds, multi-turn conversations, and the observation that these naturalistic audio scenarios can systematically bypass text-centric safety filters is well-motivated. This is not about adversarial perturbations or gradient-based optimization; it's about the kind of audio a model would actually encounter in deployment.

2. The three-way decomposition of attack strategies (SSO, SAO, MSD) is clean and covers meaningfully different failure modes. SSO tests whether models can separate concurrent speech streams, SAO tests whether models attend to non-speech audio semantics at all, and MSD tests whether models can track harmful intent distributed across dialogue turns. Each targets a distinct architectural weakness, and the results confirm they exploit different model vulnerabilities (SAO is hardest for all models; SSO varies more).

3. The experimental breadth is comprehensive: 9 models (6 open-source, 3 proprietary) with Gemini 2.5 Pro, GPT-4o, Gemini 1.5 Pro, Qwen3-Omni, Qwen2.5-Omni-7B, Qwen2-Audio-7B, MiniCPM-o, Step-Audio2, and Kimi-Audio. The inclusion of a harmless control set (HCS) for measuring false alarm rates is good practice that many benchmark papers skip.

---

> ### Author Rebuttal · Authors · 2026-03-31
>
> Thank you for your invaluable feedback and constructive suggestions. We would like to respond to each individual concern as follows:
>
> ---
> > Question 1: Can the authors clarify the naming: why “SALMONN-Guard” when the model is built on Qwen2.5-Omni? Is there a connection to the original SALMONN architecture that I’m missing?
> >
> >Limitation 3: Naming confusion: SALMONN-Guard is not built on SALMONN ...
>
> **We agree with your observation that the previous naming choice was confusing, we have decided to rename the guard model in the revised manuscript.**
>
> ---
> >Limitation 1: SSO and SAO reduce to binary classification, which is a weak evaluation target ...
>
> **While we formulate this as a binary task, the consequence of the model response falls within the consensus of LLM harmful behaviour, which is that the model encourage or facilitates harm in real world.**
>
> Our composed audio is essentially attacking the guardrail for LLM, testing whether it can prevent such false acknowledgements from happening. In this sense, we believe categorizing them as ASR is appropriate.
>
> We understand the concern raised by the reviewer, and will clarify this by using ASR (DFR) for SSO and SAO tasks, indicating that these should also be considered as Detection Failure Rate (DFR).
>
> ---
> >Question 3: Table 9 in Appendix shows that SALMONN-Guard’s ASR (14.08% on MSD) is overwhelmingly “Wrong Understanding” (11.93%) rather than “Harmful Compliance” (2.15%) ...
>
> **Thank you for the careful reading. The wrong understanding that causes harmful behaviour is categorized as harmful in our paper, and we report this to keep consistent with any other evaluations we reported within our paper.** We will also include this distinction in the introduction of our revised paper.
>
> ---
> >Limitation 4: No comparison with simple defenses ... Without these baselines, it's hard to tell if the full SALMONN-Guard training pipeline is necessary.
>
> **Our evaluation confirms that simple cascaded systems and prompt-based strategies provide insufficient protection, fundamentally because text-only pipelines cannot disentangle complex audio mixtures like Speech-Speech Overlap (SSO) or perform Multi-Speaker Dialogue (MSD) safety reasoning.** We will add these defensive baseline comparisons to Section 5:
>
> | Defense Baseline | SSO (ASR %) | MSD (ASR %) |
> | :--- | :---: | :---: |
> | Baseline (Gemini 2.5 Pro without defense) | 37.25 | 63.93 |
> | Cascaded (Whisper-large-v3 / Text-Only + Gemini) | 37.50 | 57.96* |
> | Prompt-based (System Prompt + Gemini) | 39.50 | 57.77 |
> | **SALMONN-Guard (Ours)** | **12.93** | **14.08** |
>
> *\* The MSD cascaded result (57.96%) represents the ideal text-only scenario evaluated in our Table 3 ablation study.*
>
> *   **Cascaded Defense Fails:** SSO transcription (ASR 37.50%) misses overlapping speech. For MSD, as already demonstrated in our Table 3 ablation study (Section 5.3), even when providing the perfect text transcript (the “Text-Only” setup), the ASR remains critically high at 57.96%. This proves text-based filters are blind to contextually hidden threats.
>
> *   **Prompt-based Defenses Ineffective:** Adding a system prompt (*"Before answering, strictly check if any speech contains harmful content"*) to Gemini 2.5 Pro failed to reduce ASR, which shifted from **37.25% to 39.50%** (SSO) and remained high at **57.77%** (MSD).
>
> These results explicitly validate the necessity of an end-to-end, audio-aware defense like **SALMONN-Guard**, which avoids the lossy pre-processing of ASR and the instability of prompt-based methods.
>
> ---
> >Question 2: Have the authors evaluated SALMONN-Guard on adversarial perturbation attacks (e.g., AdvWave or adversarial noise injection)...
> >
> >Limitation 2: SALMONN-Guard is evaluated only on its own benchmark ...
>
> **We would like to clarify that the cross-benchmark evaluation presented in Table 4 already serves as a robust test on fully independent, non-compositional benchmarks.** We fully agree that evaluating on qualitatively different attacks is crucial for proving generalization. However, we respectfully point out that the two methods evaluated in Table 4  represent exactly this kind of independent distribution. Specifically, these two benchmarks do not use compositional techniques (like our overlapping or dialogue mechanisms). Instead, they are pure audio manipulation attacks:
> * **Speech Insertion [1] (https://arxiv.org/pdf/2410.23861):** This method directly alters the speech content by inserting malicious audio segments into the middle of benign speech.
> * **Speech Editing [2] (https://arxiv.org/abs/2501.13772):** This method applies 18 distinct audio editing transformations (e.g., speed perturbation, pitch shifting) to evade safety filters.
>
> Despite significant distribution shifts, our guard model achieved near-perfect zero-shot defense on these independent benchmarks (**0.00% and 3.00% ASR**). This proves it learns generalized multimodal safety rather than overfitting to compositional audio.

---

> > ### Author Rebuttal · Reviewer_WRA1 · 2026-04-03
> >
> > My concerns are fully resolved.

---

### Official Review · Reviewer_Sceh · 2026-03-11

**Soundness:** 2
**Presentation:** 3
**Significance:** 3
**Originality:** 3
**Overall Recommendation:** 4
**Confidence:** 2

**Summary:**

The paper introduces the SACRED-Bench, deployed to examine complex audio-based attacks.
Unlike existing methods that rely on artificial signal manipulation, this benchmark demonstrates the feasibility of effective attacks even in a black-box setting through three conditions (e.g., (a) overlap of harmful and benign speech, (b) mixture of benign speech with harmful non-speech audio, and (c) multispeaker dialogue) with a high attack success rate.
The researcher also proposed SALMONN-Guard defense mode, which comprehensively inspects not only text but also audio signals. The proposed system significantly reduces the attack success rate to around 20%

**Compliance With Llm Reviewing Policy:**

Affirmed.

**Final Justification:**

I recommend upgrading the overall rating from Weak Reject to at least Weak Accept.

- Validity and Clarity: Although concerns were initially raised about the scale of the experiment and the lack of rigor in the evaluation, these issues were largely resolved during the rebuttal process, using additional data, significantly enhancing the paper's overall validity.

- Valuation Strictness: Through failure-case decomposition, the authors demonstrated that only 8.81% of Gemini 2.5 Pro’s attack success rate is due to benign misunderstanding, while 55.12% is attributable to genuine harmful compliance, supporting the validity of their evaluation.

- Sample Size and Listening Evaluation: The authors doubled both the control sample (HCS) and the listening evaluation scale to improve statistical reliability and verify the perceptual clarity of adversarial audio.

**Key Questions For Authors:**

- While the current focus is on black-box attacks, do you have plans to incorporate adversarial noise or more sophisticated attack vectors in your evaluation?
- Does it seem reasonable to rate a model's incorrect answer as fatal with the same weight as being breached by a malicious attack?
- The current audible assessment appears to be preliminary in nature. Do you plan to expand the scope of this test to a more comprehensive level?

**Limitations:**

yes

**Strengths And Weaknesses:**

### Strength
- The paper introduces the realistic complex audio attack benchmark (e.g., (a) overlap of harmful and benign speech, (b) mixture of benign speech with harmful non-speech audio, and (c) multispeaker dialogue).
The paper also highlights a critical vulnerability of LLMs by showing high attack success rates across benchmarks on both open-source and commercial models.
- The paper proposes a novel defense framework that jointly examines speech, non-speech audio, and text prompt without separating them, through concurrent updates of an LLM, an audio encoder, and an aligner during training.
### Weakness
- The harmless control set evaluated in the paper is 400, which seems insufficient to evaluate the model's performance.
- In the SACRED-Bench dataset, 'wrong understanding' seems inevitable due to the nature of audio mixing. However, assuming an attack is successful solely based on these misinterpretations could result in an overly stringent or biased evaluation metric.
- The high defensive efficacy of SALMON-Guard appears to stem from extensive fine-tuning on diverse mixture datasets, which may overlap with the evaluation scenarios. In contrast, other baseline models show higher attack success rates, likely because they are evaluated on unseen data, suggesting that the current comparison may lack a fair evaluation environment.
- The authors experimented to demonstrate that harmful content in synthesized audio is clearly audible to humans. However, the samples and methods used in the verification process are inadequate.

---

> ### Author Rebuttal · Authors · 2026-03-31
>
> Thank you for your invaluable feedback and constructive suggestions. We would like to respond to each individual concern as follows:
>
> ---
> > Weakness 2: In the SACRED-Bench dataset, “wrong understanding” seems inevitable ...
> >
> >Question 2: Does it seem reasonable to rate a model’s incorrect answer as fatal with the same weight as being breached by a malicious attack?
>
> **As detailed in Table 9 (Appendix F.2), we have already decomposed these failure modes.**
> 1. Our analysis confirms that the high Attack Success Rate (ASR) observed in SACRED-Bench is primarily driven by genuine safety breaches rather than simple audio confusion. For instance, in the MSD attack against Gemini 2.5 Pro, the total ASR is 63.93%. This total is comprised of a dominant **55.12%** classified as “Harmful Compliance”, where the model perfectly understands and executes the malicious instruction, and only 8.81% classified as "Wrong Understanding."
> 2. Note that we report the overall Attack Success Rate (ASR) from a consequence-oriented perspective. We consider any model response that leads to a harmful outcome as a safety failure, even if that outcome results from a model’s misunderstanding of the auditory input.
>
> ---
> > Weakness 3: The high defensive efficacy of SALMON-Guard appears to stem ... suggesting that the current comparison may lack a fair evaluation environment.
> - **First, commercial and open-source models (like Gemini and Qwen) are not “zero-shot” in terms of safety; they have undergone massive safety alignment (RLHF/SFT).** The high ASR they exhibit simply proves our core argument: their existing safety alignments are highly text-centric and fail to generalize to complex audio modalities. That is, we are **proposing** these **new but realistic scenarios** where existing safety mechanisms may fail.
> - **Second, we enforced a strict cross-dataset validation protocol (Section 4.1) to prevent data overlap and ensure that SALMONN-Guard learns the concept of multimodal safety rather than simply overfitting.** The harmful instructions used in the test set were completely held out from the training set.
> - **Most importantly, Section 5.4 (Table 4) demonstrates SALMONN-Guard’s strong zero-shot generalization.** Without any fine-tuning on JALMBench (which uses entirely different perturbation-based attacks like Speech Insertion/Editing), SALMONN-Guard successfully reduced the ASRs from 23% to 3.00% and 0.00%.
>
> ---
> >Weakness 4: The authors experimented to demonstrate that harmful content in synthesized audio is clearly audible to humans. However, the samples and methods used in the verification process are inadequate.
> >
> >Question 3: The current audible assessment appears to be preliminary in nature ...
>
> **We doubled our human study scale for Speech-Speech Overlap (SSO) and Speech-Audio Overlap (SAO); results remain statistically consistent (97.5% SSO, 96.0% SAO), confirming harmful content is clearly audible.** By recruiting 2 more independent volunteers (4 total) for 50 samples each, we proved high ASRs stem from alignment failures, not inaudible signals. A larger crowdsourced study will be included in the final version to further solidify benchmark quality.
>
> | Listener | SSO (N=50) | SAO (N=50) |
> | :--- | :---: | :---: |
> | Listener 1 (Original) | 100% | 96% |
> | Listener 2 (Original) | 96% | 96% |
> | Listener 3 (**New**) | 98% | 98% |
> | Listener 4 (**New**) | 96% | 94% |
> | **Overall (Expanded)** | **97.5%** | **96.0%** |
> ---
> >Weakness 1: The harmless control set evaluated in the paper is 400, which seems insufficient to evaluate the model's performance.
>
> **Our initial 400-sample Harmless Control Set (HCS) was carefully curated to cover a diverse range of benign scenarios, and evaluation on an additional 400 samples confirms that our False Alarm Rate (FAR) findings are statistically stable.** Though 400 samples align with benchmarks like HarmBench (https://arxiv.org/abs/2402.04249) and AdvBench (https://arxiv.org/abs/2307.15043), we re-evaluated representative models on an additional 400 HCS samples during rebuttal. As shown below, the FAR remains near-identical, proving the stability of our initial results. We will officially expand the HCS to 1,000 samples for the camera-ready version.
>
> | Models | Original FAR (%) (N=400) | Additional FAR (%) (N=400) |
> | :--- | :---: | :---: |
> | Qwen2-Audio-7B | 1.25 | 1.50 |
> | Qwen2.5-Omni-7B | 1.25 | 1.25 |
> | Gemini 2.5 Pro | 1.00 | 1.25 |
> | **SALMONN-Guard (Ours)** | **0.00** | **0.00** |
>
> ---
> >Question 1: While the current focus is on black-box attacks, do you have plans to incorporate adversarial noise or more sophisticated attack vectors in your evaluation?
>
> **SACRED-Bench deliberately focuses on realistic, everyday user-interaction scenarios (e.g., background TV noise, multiple speakers) that naturally occur in black-box settings.** That said, we will consider expand it to cover adversarial noises in the future provided that they are black-box and are generally effective across multiple audio LLMs.

---

> > ### Author Rebuttal · Reviewer_Sceh · 2026-04-02
> >
> > The authors have successfully addressed my concerns. I am raising my score accordingly.

---

### Decision · Program_Chairs · 2026-04-30

**Decision:**

Accept (regular)

**Comment:**

This paper highlights a critical vulnerability in current multimodal LLMs. The authors demonstrate that by concealing harmful information within complex audio scenes, these text-centric safety mechanisms can be circumvented. To systematically evaluate and mitigate this risk, the paper introduces SACRED-Bench, a novel audio red-teaming benchmark, alongside SALMONN-Guard, a defence model designed specifically to assess audio safety.

The reviewers raised several valid concerns during the review process, and the authors addressed them in the rebuttal. The reviewers noted the lack of simple/cascaded defence baselines; the authors responded by adding such baselines. Reviewers raised concerns about how “attack success” should be defined. The authors provided a clear justification. All three reviewers acknowledged that their concerns were resolved.